# Investigation of an Antioxidative System for Salinity Tolerance in *Oenanthe javanica*

**DOI:** 10.3390/antiox9100940

**Published:** 2020-10-01

**Authors:** Sunjeet Kumar, Gaojie Li, Jingjing Yang, Xinfang Huang, Qun Ji, Kai Zhou, Suliman Khan, Weidong Ke, Hongwei Hou

**Affiliations:** 1The State Key Laboratory of Freshwater Ecology and Biotechnology, The Key Laboratory of Aquatic Biodiversity and Conservation of Chinese Academy of Sciences, Institute of Hydrobiology, Chinese Academy of Sciences, Wuhan 430072, China; sunjeet@ihb.ac.cn (S.K.); ligaojie@ihb.ac.cn (G.L.); yangjj@ihb.ac.cn (J.Y.); suliman.khan18@mails.ucas.ac.cn (S.K.); 2College of Modern Agricultural Sciences, University of Chinese Academy of Sciences, Beijing 100049, China; 3Institute of Vegetables, Wuhan Academy of Agricultural Sciences, Wuhan 430207, China; huangxinfang99@163.com (X.H.); jiqun741@sina.com (Q.J.); zhoukai5731@163.com (K.Z.)

**Keywords:** *Oenanthe javanica*, salt stress, phenotype, ROS, antioxidants, antioxidant enzymes, transcriptome, differently expressed genes

## Abstract

Abiotic stress, such as drought and salinity, severely affect the growth and yield of many plants. *Oenanthe javanica* (commonly known as water dropwort) is an important vegetable that is grown in the saline-alkali soils of East Asia, where salinity is the limiting environmental factor. To study the defense mechanism of salt stress responses in water dropwort, we studied two water dropwort cultivars, V11E0022 and V11E0135, based on phenotypic and physiological indexes. We found that V11E0022 were tolerant to salt stress, as a result of good antioxidant defense system in the form of osmolyte (proline), antioxidants (polyphenols and flavonoids), and antioxidant enzymes (APX and CAT), which provided novel insights for salt-tolerant mechanisms. Then, a comparative transcriptomic analysis was conducted, and Gene Ontology (GO) analysis revealed that differentially expressed genes (DEGs) involved in the carbohydrate metabolic process could reduce oxidative stress and enhance energy production that can help in adaptation against salt stress. Similarly, lipid metabolic processes can also enhance tolerance against salt stress by reducing the transpiration rate, H_2_O_2_, and oxidative stress. Furthermore, the Kyoto Encyclopedia of Genes and Genomes (KEGG) pathway analysis showed that DEGs involved in hormone signals transduction pathway promoted the activities of antioxidant enzymes and reduced oxidative stress; likewise, arginine and proline metabolism, and flavonoid pathways also stimulated the biosynthesis of proline and flavonoids, respectively, in response to salt stress. Moreover, transcription factors (TFs) were also identified, which play an important role in salt stress tolerance of water dropwort. The finding of this study will be helpful for crop improvement under salt stress.

## 1. Introduction

Salinity, an abiotic stressor, has a significant impact on plant productivity [1]. A high concentration of salt leads to osmotic stress and ionic imbalance in plants, which in turn affect plant physiology, decreased biomass and biochemical processes, and ultimately lead to plant injury or death [2,3]. Until now, due to the adverse impacts of increasing salinity, approximately 1125 million hectares of land have already been affected [4,5]. In China alone, 36.7 million hectares of land including 12.3 million hectares of arable land in the north, northeast, northwest, and coastal areas has been affected by salinity [6].

To improve the efficacy of tolerance generating approaches, and increase crop yield, it is necessary to elucidate the response mechanisms of salt and unveil the underlying physiological parameters and molecular pathways. Previous physiological studies mentioned that salt stress induces oxidative stress through the formation of reactive oxygen species (ROS) in the form of superoxide (O_2_^●^^−^), hydrogen peroxide (H_2_O_2_), and hydroxyl radicals (^●^OH), which affects the cellular structures and metabolism of the plants as a result [7,8]. ROS are produced in all forms of aerobic life under stress or normal conditions. The production of ROS cause oxidative damage that has a negative impact on the role of important macromolecules [9]. Plants produce compatible osmolytes such as proline and soluble sugars, which help the plants under stress in osmotic adjustment [8,10]. Proline is produced from either ornithine or glutamate in osmotically stressed cells. It also possesses ROS scavenging activity and enhances the activity of antioxidant enzymes [11,12]. Similarly, antioxidant molecules including polyphenols and flavonoids help to minimize the negative effects of salt stress by removing free radicals, which enhances the tolerance against salt stress [13].

The oxidative damage can be also overcome by antioxidant enzymes such as ascorbate peroxidase (APX), superoxide dismutase (SOD), catalase (CAT), peroxidase (POD), and glutathione reductase (GR) [14,15]. These enzymes are present in all subcellular compartments [16]. The ascorbate–glutathione cycle is the main hydrogen peroxide-detoxification system, which is found in the chloroplast of the plants, where APX is the key enzyme [17]. The overproduction of O_2_^●^^−^ is regulated by the stromal and thylakoid-bound SODs, which convert it into H_2_O_2_, which, in turn, is converted into H_2_O by APX and CAT. POD in the cytosol and peroxisomes effectively remove H_2_O_2_ from the surrounding of the chloroplast [9,17]. Several reports mention that these antioxidant enzymes have a positive correlation with plant tolerance in drought and salt stress [15]. Higher antioxidant molecules and antioxidant enzymes activities can help plants develop stress tolerance and prevent cell death [18].

Several studies at the molecular level were also conducted to explore the mechanisms of salt tolerance in model plants, such as Arabidopsis, rice, and tobacco [19,20,21,22,23]. These studies focused on the mechanisms underlying the salt tolerance in plants from different aspects such as transcription factors (TFs), plant hormones, antiporters, and the biosynthesis of secondary metabolites [24,25,26]. However, considering the diversities among plant species, the mechanism on these model plants are limited in horticultural vegetables. One of the vegetables is the water dropwort (*Oenanthe javanica* (Blume) DC), which is a perennial aquatic vegetable, belonging to the family Apiaceae [27]. Several studies have reported that water dropwort is a rich source of dietary fibers, starch, vitamins, and minerals, with excellent medicinal properties. Hyperoside, isorhamnetin, and persicarin are the key compounds present in water dropwort, which have pharmacological activities against different ailments and are hepatoprotective, anti-inflammatory, anti-arrhythmic, and antidiabetic. These advantages have led to water dropwort being popularly cultivated in many countries, like China, Korea, Thailand, Japan, Malaysia, and Australia [27,28,29]. The total yield of water dropwort is 56.55 tons/hm^2^ in China [30]. A previous report mentioned that water dropwort is sensitive to salt stress and salinity is the main limiting factor for its production [27]. Therefore, it is necessary to explore the physiological and fundamental molecular mechanisms of salt stress tolerance in the water dropwort.

Previously, many reports provided noteworthy information to elucidate the mechanisms of salt tolerance [31,32,33]. An important considerations for current studies is to investigate unexplored mechanisms in non-sequenced plants using next generation sequencing (NGS) technology, which assists in highlighting molecular alterations and detect gene expression patterns in different plants [34,35,36,37].

However, no study has comprehensively compared the salt tolerant and sensitive cultivars in the water dropwort. Here, we firstly identified a salt tolerant [V11E0022, (Shaoyan shuiqin)] and a sensitive [V11E0135, (Hefeizhongye shuiqin)] cultivar in water dropwort. Using these cultivars, we conducted phenotypic, physiological, and high-throughput transcriptomic sequencing analysis for mechanism studies. From the present study, we identified important physiological parameters, such as enzymatic and non-enzymatic antioxidant activities, as well as potential or candidate genes involved in the salt stress response. These studies can help further elucidate the salinity response mechanisms and identify genes of interest for breeding purposes in *O. javanica*.

## 2. Materials and Methods

### 2.1. Plant Culture and Salt Treatment

Seeds of two cultivars of *Oenanthe javanica*; V11E0022 (Shaoyan shuiqin), and V11E0135 (Hefeizhongye shuiqin) were kept in wet sand for 1 month, and then the seeds were shifted to the wet filter paper and placed in the growth chamber (12/12 h) at 25 °C. The seeds germinated in 7–10 days. Seedlings were transferred to Hoagland nutrient solution [38] and grown for 14 days in the greenhouse condition with 20–25 °C and 16 h photoperiod. The composition of media was 3.59 mmol/L Ca(NO_3_)_2_, 8.7 mmol/L KNO_3_, 0.713 mmol/L N₂H₄O₃, 1.516 mmol/L MgSO_4_, 1.314 mmol/L KH_2_PO_4_, 62.5 µmol/L FeSO_4_, 44.6 µmol/L EDTA, 48.5 µmol/L H_3_BO_3_, 13.2 µmol/L MnSO_4_, 1.36 µmol/L ZnSO_4_, 0.501 µmol/L CuSO_4_, and 2.55 µmol/L (NH_4_)_2_MoO_4_. Subsequently, the plants were subjected to 0 (control), and 200 mmol/L NaCl for 7 days in Hoagland’s nutrient solution. The experiments were conducted in biological triplicates.

### 2.2. Phenotypic Parameters

After harvesting, shoots and roots were washed with distilled water and blotted dry gently with a paper towel. The fresh weight (FW), plant height, stem length, root length, number of branches, and leaves were measured.

### 2.3. Determination of Relative Water Content

The relative water content (RWC) of leaves was measured using the method described by Ghoulam et al. [39]. After recording the fresh weight (FW), leaves were immersed in distilled water inside a closed Petri dish for 4 h, and then the turgor weight (TW) of each leaf was noted. Leaf samples were then placed in a pre-heated oven at 70 °C for 24 h to obtain dry weight (DW). Afterward, RWC was calculated using the following formula:RWC% = [(FW − DW)/(TW − DW)] × 100

### 2.4. Physiological Parameters

#### 2.4.1. Determination of Lipid Peroxidation

For the determination of malondialdehyde (MDA), fresh leaves (50 mg) were homogenized with 450 µL Phosphate buffer saline (PBS) (pH 7.4, 0.1 M) using a glass homogenizer. After that, samples were centrifuged at 6800× *g* for 15 s for 3 times with the intervals of 30 s, then homogenate was again centrifuged at 2500× *g* for 10 min. After centrifugation, the supernatant was used for the MDA analysis using a commercially available test kit (Nanjing Jiancheng Bioengineering Institute, Nanjing, China), and finally the absorbance was measured at 530 nm [40].

#### 2.4.2. Assays for Hydrogen Peroxide, GSH, and Antioxidant Enzymes

For the determination of H_2_O_2,_ GSH, and antioxidant enzymes, 100 mg fresh leaf samples were homogenized with 900 µL of PBS (pH 7.4, 0.1 M) using a glass homogenizer. Afterward, they were centrifuged at 3500× *g* for 12 min. A supernatant was used for the determination of H_2_O_2,_ GSH, and antioxidant enzyme activities including APX, SOD, POD, and CAT using commercially available test kits (Nanjing Jiancheng Bioengineering Institute, Nanjing, China) [40,41,42,43].

H_2_O_2_ forms a complex with molybdate that can be measured by absorbance at 405 nm. The GSH content was determined using a Glutathione Assay Kit (A006-1, Nanjing Jiancheng Bioengineering Institute) for reduced glutathione that followed the DTNB [5,5′-dithiobis (2-nitrobenzoic acid)] method. The absorbance was measured at 420 nm and GSH content was expressed as mg g^−1^ protein [40,41,42,43].

The activity of APX was determined using an APX Assay Kit (A123-1-1) obtained from Nanjing Jiancheng Bioengineering Institute. APX catalyzed the oxidation of ascorbate at 290 nm using a spectrophotometer and was expressed as U mg^−1^ FW. One unit activity of APX is the amount of enzyme which oxidizes 1 μmol ascorbate per min in 1 mg fresh sample [44]. The activity of SOD was determined by using SOD Assay Kit (A001-1) obtained from Nanjing Jiancheng Bioengineering Institute and was presented as U mg^−1^ FW. One unit of SOD activity is the amount of extract that gives 50% inhibition in the reduction of xanthine as monitored at 550 nm [45]. The activity of POD was measured by using the POD Assay Ki (A084-3-1, Nanjing Jiancheng Bioengineering Institute, Nanjing, China) on the basis of guaiacol oxidation at 470 nm by H_2_O_2_ and expressed as U mg^−1^. The change in absorbance at 470 nm was recorded for every 20 s by a spectrophotometer [46]. One unit of POD activity is the amount of enzyme, which causes the decomposition of 1 μg substrate per minute in 1 mg fresh sample at 37 °C. Similarly, the activity of CAT was measured by using CAT Assay Kit (A007-1, Nanjing Jiancheng Bioengineering Institute), and was demonstrated as U mg^−1^ FW. One unit of CAT activity is the amount of enzyme, which causes the decomposition of 1 μmol H_2_O_2_ per minute in 1 mg fresh sample at 37 °C [47].

#### 2.4.3. Determination of Proline and Soluble Sugars

To determine proline content and soluble sugars, 50 mg fresh samples of leaves were homogenized according to the instruction manual provided by the company (Nanjing Jiancheng Bioengineering Institute, Nanjing, China), and the absorbance were measured at 520 nm and 620 nm for the proline and soluble sugars, respectively [40].

#### 2.4.4. Determination of Total Polyphenols and Flavonoids Content

For the determination of total polyphenols, the Folin–Ciocalteu method was used with some modifications given by Alves et al. [48]. Accordingly, 100 µL of fresh leaves were homogenized and extracted by using 10% ethanol. 500 µL sample were mixed with 2.5 mL Folin–Ciocalteu reagent (1:10), then 2 mL 0.75 g/mL Na_2_CO_3_.10H_2_O solution was added. After incubation at 45 °C for 15 min, the mixture was placed at room temperature for 30 min. Finally, absorbance level was measured at the wavelength of 765 nm, and the results were analyzed against standard gallic acid (GAE/g).

For the determination of flavonoids, aluminum chloride method was used described by Barroso et al. [49]. For this purpose, 4 mL distilled water was mixed with 300 µL NaNO_2_ solution (0.5 g/mL), then 1 mL of plant extract was added. After 5 min, 300 µL AlCl_3_ (1 g/mL) solution, 2 mL NaOH (1 mol/L), and 2.4 mL double distilled water were added simultaneously. Finally, the absorbance level was measured at the wavelength of 510 nm, and the results were analyzed against standard catechin (CE/g).

#### 2.4.5. Ions Determination

For determination of ions (Na^+^ and K^+^ contents), 100 mg of dried samples of leaves were digested with 6 mL nitric acid using a microwave digestion system (Multiwave 3000, Anton Paar, Austria) for 1.5 h. Digested samples were diluted up to 10 mL with ultra-deionized water. Ions were determined by inductively coupled plasma-atomic emission spectroscopy ICP-OES (Optima 8000, PerkinElmer, Waltham, MA, USA) at the public technical service center in the Institute of Hydrobiology, Chinese Academy of Sciences, Wuhan, China [50].

### 2.5. Statistical Analysis

All data in triplicates were analyzed statistically by SPSS 25.0 and subjected to independent *t*-tests after the determination of homogeneity of variances by Levene’s test. The significant difference level was set at *p* < 0.05, and all the data were represented as the mean ± standard deviation (S.D).

### 2.6. Transcriptomic Analysis

#### 2.6.1. RNA Quantification and Qualification

The total RNA from the control and NaCl-treated leaves was extracted using Trizol (Invitrogen, Santa Clara, CA, USA) following the manufacturer’s instructions. The purity and concentration of RNA samples were tested using BioDrop uLite (80-3006-51) to ensure the use of qualified samples for transcriptome sequencing.

#### 2.6.2. Illumina Library Construction and Sequencing

A total amount of 1 μg RNA per sample was used as the input material for the RNA sample preparations. Sequencing libraries were generated using NEBNext^®^Ultra™ RNA Library Prep Kit for Illumina^®^ (NEB, Ipswich, MA, USA) following the manufacturer’s recommendations and index codes were added to attribute sequences to each sample. Briefly, mRNA was purified from total RNA using poly-T oligo-attached magnetic beads. Fragmentation was carried out using divalent cations under an elevated temperature in NEBNext First-strand synthesis reaction buffer (5X). First-strand cDNA was synthesized using random hexamer primer and M-MuLV Reverse transcriptase. Second strand cDNA synthesis was subsequently performed using DNA Polymerase I and RNase H. The remaining overhangs were converted into blunt ends via exonuclease/polymerase activities. After adenylation of 3’ ends of DNA fragments, NEBNext Adaptor with hairpin loop structure were ligated to prepare for hybridization. In order to select cDNA fragments of preferentially 240 bp in length, the library fragments were purified with AMPure XP system (Beckman Coulter, Beverly, CA, USA). Then, 3 µL USER Enzyme (NEB, Ipswich, MA, USA) was used with size-selected, adaptor-ligated cDNA at 37 °C for 15 min followed by 5 min at 95 °C before PCR. After this, PCR was performed with Phusion High-Fidelity DNA polymerase, Universal PCR primers, and Index (X) Primer. At last, PCR products were purified (AMPure XP system) and library quality was assessed on the Agilent Bioanalyzer 2100 system [51].

The clustering of the index-coded samples was performed on a cBot Cluster Generation System using TruSeq PE Cluster Kit v3-cBot-HS (Illumia) according to the manufacturer’s instructions. After cluster generation, the library preparations were sequenced on an Illumina Hiseq 2000 platform and paired-end reads were generated [51].

#### 2.6.3. Quality Control and Transcriptome Assembling

Raw data (raw reads) of FASTQ format were firstly processed through in-house Perl scripts. In this step, clean data (clean reads) were obtained by removing reads containing adapter, ploy-N and low-quality reads from raw data. At the same time, Q20, Q30, GC-content, and sequence duplication levels of the clean data were calculated. All the downstream analyses were based on clean data with high quality. The filtered high-quality reads were used for transcriptome assembling by the Trinity software with default parameters [52]. The clean datasets of the samples were pooled for de novo assembling and comprehensive sequence library construction [51].

Transcriptomic data were submitted to the NCBI Sequence Read Archive (SRA) database. SRA accession number: PRJNA647709.

#### 2.6.4. Identification of Differentially Expressed Genes (DEGs)

Gene expression levels were estimated by RSEM for each sample. Clean data were mapped back onto the assembled transcriptome, and read count for each gene was obtained from the mapping results [53]. The differential expression analysis of two conditions/groups was performed using the DESeq R package (1.10.1). The resulting *p*-values were adjusted using the Benjamini and Hochberg’s approach for controlling the false discovery rate. Genes with an adjusted *p*-value < 0.05 found by DESeq were assigned as differentially expressed. Log2 (fold change) >1 was set as the threshold for significantly differential expression [51].

#### 2.6.5. Functional Annotation

Gene Ontology (GO) enrichment analysis of the differentially expressed genes (DEGs) was implemented by the topGO R packages-based Kolmogorov–Smirnov test. KEGG database was used to identify the genes with their putative pathways [54]. We used KOBAS software to test the statistical enrichment of differential expression genes in KEGG pathways [55]. Differentially expressed transcription factors were then identified by alignment to the Plant Transcription Factor Database PlnTFDB (http://plntfdb.bio.uni-potsdam.de/v3.0/) and PlantTFDB (http://planttfdb.cbi.pku.edu.cn) [51].

#### 2.6.6. Validation of DEGs Using Quantitative Real-Time PCR (qRT-PCR)

The total RNA from the control and NaCl-treated leaves was extracted using Trizol (Invitrogen, Santa Clara, CA, USA), following the manufacturer’s instructions. Reverse transcription reactions were performed using SuperScript III reverse transcriptase (Invitrogen, Grand Island, NY, USA), according to the manufacturer’s instructions. RT-qPCR assays were performed on a CFX96 Real-time PCR system, Bio-rad, using the SYBR Premix Ex Taq™ II (Tli RNaseH Plus) (TaKaRa Biotech. Co., Kusatsu, Shiga, Japan). Primers were designed using Primer3 software. The primer pairs for the amplification of the 10 candidate genes are listed in Appendix A. The following amplification protocol was used: denaturation at 95 °C for 30 s, 40 cycles of 95 °C for 5 s, 55 °C for 30 s, and 72 °C for 30 s), and a final extension step. The melting curve was obtained by heating the amplicon from 65 °C to 95 °C at 0.5 °C s^−1^. The relative transcript abundance was based on the mean of three biological replicates at each sampling time point using the 2^−ΔΔCT^ approach [56]. *Actin* (*c55427.graph_c0*) gene was used as the reference gene in the study.

## 3. Results

### 3.1. Effect of Salt Stress on Growth, Biomass, and Relative Water Content of Water Dropwort

To detect whether plant growth properties were affected by salinity in both treated cultivars, we compared vegetative parameters of two water dropwort cultivars and found that plant height, stem length, root length, number of branches, number of leaves, and shoot and root fresh biomass in the treatments were significantly lower than the control (Figure 1). Following high salinity exposure, we found that V11E0135 is sensitive and V11E0022 is tolerant to salt stress. V11E0135 showed more reduction in studied plant growth parameters and biomass as compared to the V11E0022 (Table 1). Relative water content (RWC) is considered as an accurate and easy parameter to check salt and drought stress [57]. Similarly, more reduction in the RWC was found in V11E0135 in comparison to the V11E0022 cultivar, under salt stress treatment (Figure 2), indicating the different responses between these sensitive and tolerant cultivars.

### 3.2. Antioxidant Defense Systems Are Involved in the Salinity Response of Water Dropwort

To further analyze the different responses, we detected antioxidant defense systems between these cultivars.

#### 3.2.1. ROS Content and Lipid Peroxidation under Salt Stress

We found that salt stress significantly induced lipid peroxidation in terms of malondialdehyde (MDA) content in the leaves of both water dropwort cultivars (*p* < 0.05). Moreover, a high MDA content was present in V11E0135 compared to V11E0022 (Figure 3A). Similarly, H_2_O_2_ production rate was also significantly elevated in the leaves of both cultivars as compared to the control (*p* < 0.05). Moreover, significantly much higher H_2_O_2_ content was observed in V11E0135 compared to V11E0022 (*p* < 0.05) (Figure 3B).

#### 3.2.2. Effects of Salt Stress on Osmolytes and Antioxidant Molecules

Compatible osmotyles such as proline, soluble sugars, and antioxidant molecules including polyphenols, flavonoids, and glutathione help in osmotic adjustment, and reduce the negative effects of salt stress by removing free radicals. Proline and total polyphenol concentrations were found to be higher in both cultivars under salt stress in comparison to the control (*p* < 0.05). However, V11E0022 showed a significantly higher level of proline, total polyphenolic compounds, and flavonoids compared to their counterpart (Figure 4A,C,D). Soluble sugars were also insignificantly increased in V11E0022 (Figure 4B). On the other hand, V11E0135 showed an insignificant decline in soluble sugars and an insignificant increment in the concentration of polyphenols under salt stress compared to the control (Figure 4B,C). Interestingly, GSH concentration was found higher in V11E0135 as compared to its other counterparts (Figure 4E).

#### 3.2.3. Effects of Salt Stress on Antioxidant Enzymes Activities

Antioxidant enzymes (APX, SOD, POD, and CAT) were also found higher in V11E0022 under salt stress treatment. Notably, a significant increase was observed in the activities of APX and POD (*p* < 0.05) (Figure 5A,B). On the other hand, the activities of APX and CAT were significantly decreased in V11E0135 compared to the control (*p* < 0.05) (Figure 5A,D). Whereas, an insignificant decline was observed in the activities of POD and SOD in V11E0135 under salt stress treatment (Figure 5B,C).

### 3.3. Effect of Salinity on Ionic Content

We detected the ionic content of the water dropwort under salt stress. NaCl stress significantly enhanced the Na^+^ content in the leaves of both cultivars (*p* < 0.05). Leaves of V11E0135 showed more uptake of Na^+^ ion compared to its counterparts (Table 2). On the other hand, K^+^ content in the leaves of both cultivars decreased with the increase of salt stress. However, V11E0135 at 200 mmol/L NaCl showed a more decrease in uptake of K^+^ in comparison to V11E0022 (Table 2).

### 3.4. Transcriptomic Analysis Revealed Potential Mechanism for Salt Tolerant in Water Dropwort

#### 3.4.1. Transcriptomic Sequencing and Assembly

Illumina HiSeq high throughput sequencing platform was used for the comparative analysis of two water dropwort cultivars, V11E0022 and V11E0135, cultivars under control and salt treatment. Total 12 samples were sequenced, the high-quality pair end reads were obtained in three control and three NaCl treated samples of V11E0135 and V11E0022, respectively. The percentage of high-quality score (Q30) was more than 93.50%, GC content was above 43.9%, and the mapped ratio was more than 76%. Furthermore, total of 58,126 unigenes and 140,405 transcripts were generated from de novo transcriptomic assembly (Appendix A).

#### 3.4.2. Identification of Differentially Expressed Genes (DEGs) under Salt Stress

We carried out an evaluation of differentially expressed genes (DEGs) in the leaves of both water dropwort cultivars under control and salt stress (200 mmol/L). Data revealed that V11E0135 has total 1579 DEGs, in which 732 were upregulated and 847 downregulated, whereas, V11E0022 showed 1901 DEGs, 680 of which were upregulated and 1221 of which were downregulated (Appendix A). In comparative transcriptome analysis, we found 579 common DEGs in V11E0135 and V11E0022 (Figure 6A). Overall, DEGs profiles were divided into 4 groups based on overexpression (Figure 6B), which showed that group D was found to be upregulated in the control of both cultivars. Group C was found to be upregulated only in the control and treated groups of the V11E0135 cultivar, Group B showed higher upregulation in salt treated groups of both cultivars. However, Group A was found to be comparatively more upregulated in the V11E0022 cultivar compared to its counterparts under salt treatment.

#### 3.4.3. Functional Annotation of the DEGs

For function elucidation, DEGs were BLAST against KOG, GO, KEGG, COG, Pfam, Swiss-Prot, eggNOG, and Nr databases. Total 1777 and 1482 DEGs were annotated in V11E0022 and V11E0135, respectively against public databases under salt stress response (Appendix A).

##### GO Biological Processes

GO enrichment analysis of DEGs revealed that total 831 and 648 biological processes terms were enriched in V11E0022 and V11E0135, respectively. Among this, 297 were found to be upregulated in V11E0022 and 283 in V11E0135. Appendix A included the significantly enriched (upregulated) in V11E0022 compared to V11E0135 under salt stress. A few important biological processes terms involved in the salt stress response including the lipid metabolic process, glycolytic process, and response to salt stress were also present (Appendix A).

In the present study, 10 genes involved in carbohydrate metabolisms and 4 genes of glycolytic processes were overexpressed in V11E0022, such as *Pyruvate kinase* (*c78535.graph_c0*) and *ENO1* (*c81985I.graph_c0*) genes of the ‘glycolytic process’. Likewise, *GAPC* (*c85818.graph_c2*) and *HXK1* (*c82835.graph_c0*) were also found overexpressed under salt stress (Figure 7; Appendix A).

In present study, we found nine overexpressed DEGs in V11E0022, involved in lipid metabolic process, such as, *c92298.graph_c1* (*PDAT2*), *c77165.graph_c0* (*CSE*), *c70041.graph_c0* (*FAD2*), and *c86738.graph_c1* (*APA1*), compared to V11E0135 (Figure 7; Appendix A).

##### KEGG Pathway Enrichment Analysis

The KEGG (Kyoto Encyclopedia of Genes and Genomes) database was used to investigate the enriched pathways. KEGG enrichment pathway analysis of DEGs revealed that 374 DEGs were allotted KEGG IDs and were categorized into 106 pathways in V11E0022 under salt stress response. Whereas, 365 DEGs were allocated KEGG IDs and categorized into 102 pathways in V11E0135. Furthermore, 128 DEGs were found to be upregulated in 85 pathways of V11E0022 under salt stress, while, 206 upregulated DEGs were present in 86 pathways of V11E0135 (Appendix A).

Total 14 DEGs of plant hormone signal transduction pathway were found comparatively overexpressed in V11E0022 compared to its counterpart (Figure 7; Appendix A). Furthermore, in arginine and proline metabolic pathway, four upregulated DEGs were present in V11E0022 (Figure 7). Similarly, four DEGs involved in flavonoid, flavone and flavonol biosynthesis pathways were found to be upregulated in V11E0022 under salt stress (Figure 7).

##### Differentially Expressed Transcription Factors

The current study showed that 138 differentially expressed TFs were identified in V11E0022 (56 genes upregulated and 82 downregulated) under salt stress. These were further classified into different 28 families, and the most enriched families were zinc finger-TFs that have 8 upregulated/14 downregulated DEGs, followed by bHLH have 8 upregulated/11 downregulated, MYB have 5/8, bZIP have 7/3, ERF have 2/4, and WRKY have 3/3 DEGs. Moreover, 124 differentially expressed TFs were found in V11E0135 (49 genes upregulated and 75 downregulated), and these were further classified into 26 families. Among these most enriched families were zinc finger-TFs, which have 16 upregulated/10 downregulated DEGs, followed by bHLH have 5 upregulated/7 downregulated, WRKY have 2/8, MYB have 3/6, bZIP have 3/3 DEGs, and ERF have 8 downregulated (Figure 8; Appendix A).

### 3.5. Validation of DEGs

For validation of the RNA-Seq data, quantitative reverse transcription PCR (qRT-PCR) analysis was performed with 10 candidate DEGs. Salt treated and control samples from both cultivars were used for qRT-PCR. Results showed that the expression profiles of selected genes under qRT-PCR were in agreement with the results obtained from the RNA-Seq, which infers its reproducibility and accuracy (Figure 9).

## 4. Discussion

Salt stress causes osmotic stress, and induces ionic and nutrient imbalance which could negatively affect different mechanisms of plant growth and development [3]. The current study showed the effects of different concentrations of NaCl (0 and 200 mmol/L) on the growth of two water dropwort cultivars. This study proposed the tolerant cultivar based on its performance against salt stress, and also suggested some physiological parameters as well as molecular mechanisms to evaluate the salt tolerance in the water dropwort.

### 4.1. Phenotypic and Physiological Studies Identified Tolerant and Sensitive Cultivars of Water Dropwort under Salt Stress

It has been investigated widely that NaCl accumulation in the cell wall and cytoplasm can reduce plant length as well as the number of branches and leaves [58,59,60,61,62]. Current studies also showed that plant length, as well as the number of branches and leaves were more affected in V11E0135 compared to its counterpart. Moreover, due to salt stress, the fresh biomass of shoot and root in V11E0135 showed more reduction compared to V11E0022. This decrease of growth and biomass is a common effect of salt stress on plants [63]. Inal et al., in a study on *Daucus carota,* also found the reduction of fresh biomass under salt stress [64]. According to Meriem et al., more decline in sensitive cultivars compared to tolerant was observed in coriander under all treatments [61].

In our current study, we found that RWC was decreased in both cultivars under salt stress, notably more reduction in V11E0135. As the water potential of salt in the soil can negatively affect the osmotic potential of dissolved solutions, this negativity will proliferate with the increase of soluble substances. Therefore, this increase of Na^+^ ion inside the cell organelles causes membrane leakage and structural changes. RWC is an accurate and simple parameter to confirm salt and drought stress in the plants [57]. Various reports published on fennel, strawberry, and *Vicia faba*, also support our findings, showed a reduction in the RWC under salt treatment, and relatively more reduction was detected in sensitive cultivars [65,66,67].

Salt tolerant cultivars with their predominant scavenging capacity exhibit less lipid peroxidation and ROS production, such as H_2_O_2_ compared to their sensitive counterparts [68]. The present study resulted in significantly higher MDA and H_2_O_2_ concentrations in V11E0135 compared to V11E0022 under salt stress. Shafeiee and Ehsanzadeh on fennel [65], Verma and Mishra on mustard [69], and Wang et al. on alfalfa [15] also reported the higher MDA and H_2_O_2_ level in sensitive cultivars as compared to their tolerant counterparts under salt stress. From the results of this study, we might deduce that due to the lipid peroxidation and H_2_O_2_ accumulation, the phenotype of V11E0135 was more affected, compared to the V11E0022. To overcome the oxidative damage, plants have a defense system in the form of osmoprotectants, non-enzymatic compounds, and antioxidant enzymes, which are always sharply induced under abiotic stress conditions [8,10,14,15,70,71]. In the current study, proline concentration was found to be significantly higher in V11E0022 compared to its counterpart. Similar results were also reported by many researchers in coriander, tobacco, and canola plants. Proline as a compatible solute accumulates in the plants under salt stress and performs its osmoprotectant activity. It also helps to protect the enzymes, stabilize their structures, and acts as an ROS scavenger [10,67]. The higher RWC in V11E0022 is due to these osmolytes. Tolerant cultivars retain more water due to proline and sugars. Similarly, soluble sugars were also found to be comparatively higher in V11E0022. This increase could be due to an increase in enzymatic activities in plants which in turn support the cells to stabilize their structure and function by interacting with macromolecules [57,72]. Previously, soluble sugars were also found higher in different plants such as coriander and canola under salt stress [57,61].

Total polyphenols not only induce activities of antioxidant enzymes, but also assist as antioxidants themselves under abiotic stress conditions [73,74]. In the present study, the content of total polyphenols and flavonoids increased significantly in V11E002 compared to its counterparts. Similarly, a higher production of total polyphenols and flavonoids was found in *Cichorium spinosum*, *Amaranthus tricolor*, *Fagopyrum esculentum*, and *Portulaca oleracea* in comparison to their control conditions [75,76,77,78]. We suggest that total polyphenolic compounds and flavonoids noticeably contributed to the defense in water dropwort under salt stress. This increment of total polyphenolic compounds and flavonoids under salt stress may be due to the rise in important phenolic compounds such as gallic acid, salisylic acid, p-hydroxybenzoic acid, vanilic acid, trans-cinnamic acid, chlorogenic acid, iso-quercetin, m-coumaric acid, and rutin [79]. Antioxidant GSH participates in several cellular processes under abiotic stress; it acts as a substrate for glutathione S-transferase and glutathione peroxidase. Furthermore, GSH can directly detoxify O_2_^−^ and OH^−^, and therefore contribute to ROS scavenging [42]. Previous studies showed that biosynthesis and accumulation of cellular GSH can enhance tolerance in crops in biotic and abiotic stress conditions [80]. In the current study, we found a higher GSH level in both cultivars under salt stress in comparison to the control condition, interestingly, V11E0135 showed higher GSH compared to V11E0022, and it might be due to the respiration that plays an important role in the GSH synthesis. Several metabolites are produced during respiration such as glycine, which might be used for the synthesis of GSH [81]. This higher GSH content is concomitant with a higher respiration rate in V11E0135.

Furthermore, salt tolerant cultivars possess more enzymes activities and compared to sensitive ones [70,71]. SOD is the first line of a defense mechanism which converts the O_2_^−^ into H_2_O_2_, and this H_2_O_2_ is then converted into water and oxygen by CAT, similarly, APX also helps to reduce H_2_O_2_ [71,82]. POD also helps to efficiently scavenge H_2_O_2_ [71]. Moreover, these SOD, POD, CAT, and APX activities were found higher under salt stress as reported in tomato plant [83] and pepper plant [84].

The results of phenotypic (growth, biomass, and RWC) and physiological studies (ionic content and antioxidant defense system) showed that V11E0022 is more tolerant than V11E0135. This provides us the background to elaborate the mechanisms involved in the salt stress response of water dropwort. Based on all phenotypic results of the current study, it is suggested that the tolerance of the water dropwort may due to higher concentration of proline, antioxidants including polyphenols and flavonoids, and better antioxidant enzymes activities viz SOD, POD, CAT, and APX, and the lower H_2_O_2_ and lipid peroxidation level in the leaves. It was also suggested that proline, flavonoids, APX, and CAT could play an efficient role in the tolerance of water dropwort against salt stress in comparison to other studied parameters.

### 4.2. Transcriptomic Analysis Results Revealed the Molecular Mechanism under Salt Stress in Water Dropwort

Transcriptomic analyses were performed for two water dropwort cultivars under salt stress conditions, and expression profiles were compared between the control and salt treated plants.

Lipids are an important component of membranes, which assists in defense by acting as signal mediators to protect plants under biotic and abiotic stress [85]. These lipids also act as mitigators, which reduce the transpiration rate by increasing the hydrocarbon layers in aerial parts of plants in response to drought and salt stress [85,86,87,88]. In this study, we found nine overexpressed DEGs in V11E0022, involved in lipid metabolic process, such as *CSE*, *PDAT2*, *FAD2*, and *APA1*. *CSE* was found to be involved in lignin biosynthesis and its response to cadmium and zinc stress as well as H_2_O_2_ and oxidative stress [89,90]. Previous studies showed that genes of phospholipid diacylglycerol acyltransferase (PDAT) including, *CsPDAT1* and *CsPDAT2* were found to be significantly upregulated in leaves under salt stress, and higher biosynthesis of triacylglycerol (TAG) was also observed [91]. Phytohormone based transcription factor induces the expression of PDAT in vegetative tissues, which helps in TAG biosynthesis. Stress-induced ABA accumulation and TAG biosynthesis are linked with each other [91]. Similarly, Dar et al. [92] reported that *FAD2* have enhanced salt stress tolerance by increasing the dienoic fatty acids content and played an important role in regulating and maintaining Na^+^/H^+^ antiporters. Moreover, Sebastián et al. [93] mentioned that the overexpression of aspartic protease gene (*APA1*) helps to improve drought tolerance by minimizing the stomatal index, stomatal density, and stomatal aperture, also mentioned that ABA generated under drought induce *APA1* expression, which further induces the expression of different genes. In our physiological studies, we found less reduction in the RWC of V11E0022, which is indicative of a lower transpiration rate during salt stress conditions. From all these reports, we can assume that the upregulation of DEGs involved in the lipid metabolic process enhanced salt stress tolerance in water dropwort by reducing the transpiration rate, H_2_O_2_, and oxidative stress.

In the present study, a total of 14 upregulated DEGs in plant hormone signal transduction pathways, such as ABA, ethylene, and transcription regulatory elements related to stress response were identified in V11E0022. For example, serine/threonine-protein kinase *SAPK2* (*c85598.graph_c0*), protein ethylene insensitive 3 *EIN3* (*c87915.graph_c0*), and gibberellin receptor *GID1B* (*c82782.graph_c0* and *c88437.graph_c0*) were observed to be upregulated under the NaCl treated samples of V11E0022. Previously conducted studies also showed the upregulation of *SAPK2*, *EIN3*, and *GID1B* in the leaves of rice, tea plant, and Arabidopsis, respectively, under salt and drought stress [94,95,96]. The overexpression of *SAPK2* supported rice to stand against salt stress. It promoted the production of osmotically active metabolites and ROS scavengers, as well as enhanced the expression of CAT and SOD [95]. Our physiological studies also showed higher activities of SOD and CAT in the V11E0022 cultivar under salt stress might to be due to this gene. Similarly, *EIN3* enhanced tolerance in Arabidopsis by increasing the activity of POD and decreasing the accumulation of ROS (H_2_O_2_) under salt stress [97]. In current physiological studies, the higher POD activity and lower ROS accumulation detected in the V11E0022 cultivar under salt stress might be due to this gene.

Flavonoids are an essential type of secondary metabolites, which play a vital role in plants against oxidative damage caused by ROS and MDA during abiotic stress [98], which help to enhance salt tolerance when accumulated [99,100]. In the current study, we observed the upregulation of four DEGs involved in flavonoid, flavone, and flavonol biosynthesis (*c82620.graph_c0*, *c86889.graph_c0*, *c87219.graph_c0*, and *c67471.graph_c0*) in V11E0022 under salt stress. Previously conducted studies also showed that salt stress induced the flavonoids accumulation in plants and enhanced their quality [98,101]. The physiological results of the current study also showed higher content of flavonoid in V11E0022 under salt treatment in comparison to other counterparts, and this is due to the overexpression of the genes involved in the flavonoid biosynthesis pathway.

Similarly, proline plays an important role to stand against osmotic and salt stress [102]. The present study showed that genes involved in salt stress tolerance pathways were found higher in V11E0022, such as arginine and proline metabolism. Physiological studies also showed that the content of proline in V11E0022 was significantly higher in comparison to the V11E0135 under salt stress, which supports the reliability of our RNA-Seq results.

TFs play a crucial role in regulating genetic expression under abiotic stresses such as drought, salinity, and cold [103]. Numerous TFs including MYB, bZIP, WRKY, bHLH, ZF, NAC, and AP2/ERF families related to salt stress have been previously identified in plants [23,103,104,105,106,107,108,109]. In the current study, we found a higher number of upregulated TFs such as BHLH, bZIP, MYB, HSF, GRAS, and ERF in V11E0022 cultivar of water dropwort compared to its counterpart under salt stress.

Previous reports showed that some bZIP, and bHLH-TFs were found linked with the ABA signal pathway that regulates the important cell processes in response to salt stress [110,111]. In the present study, we also found some upregulated TFs in V11E0022, including *c90670.graph_c1* (*ABF2*) and *c72446.graph_c1* (*ABF3*) involved in ABA signaling pathways under salt stress. Zandkarimi et al., on grapes [112], and Chang et al., on Arabidopsis [113], also reported on the role of the *ABF* genes in salt tolerance, which induce the expression of ROS scavenging genes. Moreover, other TFs also play a significant role in response to salt stress, such as WRKY-TFs regulates the gene expression under salt stress [114]. *WRKY3* is a good regulator that reduces oxidative stress and respond in multiple ways for salt stress tolerance in the tomato [115]; similarly, *WRKY7* can also be found to be upregulated under salt stress in the potato [116]. We also observed the upregulation of *c86734.graph_c0* (*WRKY3*), *c89655.graph_c0* (*WRKY7*) genes in the leaves of V11E0022 under salt stress.

Similarly, the MYB TF family is also important for salt stress tolerance [117]. Some previously reported MYB TFs related to abiotic stress were found to be upregulated in the present study. These genes include *c76821.graph_c0* (*MYB2*), *c83141.graph_c0* (*MYB44*), *c81261.graph_c1* (*MYB48*), and *c86802.graph_c0* (*SRM1*), which can perform an important role in tolerance of water dropwort against salt stress (Appendix A). Previous reports also showed that these genes have enhanced tolerance against salt stress [109,118,119,120]. Furthermore, heat shock factors (HSFs) also play an essential role in response to biotic and abiotic stresses [121], such as *HSFA1s* previously showed tolerance to salinity, osmotic, and oxidative stresses in Arabidopsis [122]. The current study also showed the upregulation of HSFs such as *c80282.graph_c0* (*HSFB2A*), *c85160.graph_c0* (*HSFA2B*), and *c82901.graph_c0* (*HSFA4B*) under salt stress in V11E0022. These TFs can play a role in response to salt stress in water dropwort.

Glycolysis plays an important role in the development of plants and helps them to adapt against abiotic stresses through the generation of ATP and carbohydrates metabolites. In 2019, Yassin et al. stated that the low uptake of Na^+^ and higher uptake of K^+^ in higher plants are the main indicator for salt tolerance in higher plants [68]. In the current study, high Na^+^ uptake and comparatively low uptake of K^+^ were observed in the V11E0135 cultivar. The same pattern of results has been reported in coriander, celery, and wheat [59,61,68]. Transcriptomic results of the current study also showed that the expression of five genes related to *NHX1* and *NHX2* exchangers, and two genes involve in hydrogen ion transmembrane such as *ATPase 4* and *KEA5* were significantly higher in V11E0022 as compared to V11E0135. A previous study showed that NHX1 and NHX2 ion exchangers are the important players in responses to salinity, and are involved in K^+^ uptake to maintain homeostasis, stomatal function, and the regulation of pH in plants [123,124,125]. The results of these parameters suggested that a low level of Na^+^ uptake in V11E0022 could help the plant tolerate salt stress.

It was previously reported that the overexpression of *HXK1* increases tolerance by regulating *NHX1* gene expression to salt stress in *Malus domestica*, similarly, *GAPC* helps in NaCl stress tolerance by acting as sensor for H_2_O_2_ in *Leymus chinensis* [126,127]. Reports showed that the carbohydrates are not only used to adjust the osmotic balance but also play an important role as protective agents in plants to regulate homeostasis under salt stress [51]. We found the upregulation of *GAPC* (*c85818.graph_c2*) and *HXK1* (*c82835.graph_c0*) in the current study, similar to previous reports. We also found overexpression of *PK* and *ENO1* genes, instead of *ENO2* in the current study, which is also supported by previous reports. Previous studies showed the interaction of *PK* and *ENO2* in response to salt stress has reduced the level of H_2_O_2_, and enhanced the growth and development in Arabidopsis [128], suggesting that water dropwort may increase the metabolism of glycolysis to reduce the accumulation of carbohydrate in the cells. In the current study, the overexpression of the previously mentioned genes in the water dropwort may function together to reduce the oxidative stress and help to enhance the respiratory metabolism and mitochondrial electron transport, which consequently minimizes the effect caused by salt stress damage.

We did not find genes related to antioxidant enzymes activity among the differentially expressed genes. However, based on the enzymatic activity tests, we suggest that post-translational modifications, protein-protein interactions, and phenotype may have influenced the activities of the enzymes under salt stress [129].

## 5. Conclusions

In conclusion, based on phenotypic and physiological results, it is demonstrated that the V11E0022 cultivar has potential tolerance against salt stress. Moreover, antioxidant enzymes (POD, APX, and CAT), osmolyte and antioxidant molecules (proline, polyphenols, and flavonoids) played a significant role in response to salt stress tolerance in water dropwort. It was also suggested that proline, flavonoids, APX, and CAT could play an efficient role in the tolerance of water dropwort against salt stress in comparison to other studied parameters. These studies draw a distinct line between sensitive and tolerant cultivars of the water dropwort, which provides the base to understand the mechanisms on salinity research. Furthermore, transcriptomic data revealed that many DEGs involved in antioxidants including lipid metabolic process, carbohydrate metabolism, plant hormone signal transduction, arginine and proline metabolism, secondary metabolites pathways, and TFs were identified in response to salt stress. This study will provide new insights to better understand the physiological and molecular mechanism underlying salt stress in the water dropwort and assist in breeding programs under abiotic stress. Overall, the genes identified in this study can be used as candidates for genetic modifications in salt sensitive cultivars of the water dropwort and related horticultural vegetables.

## Figures and Tables

**Figure 1 antioxidants-09-00940-f001:**
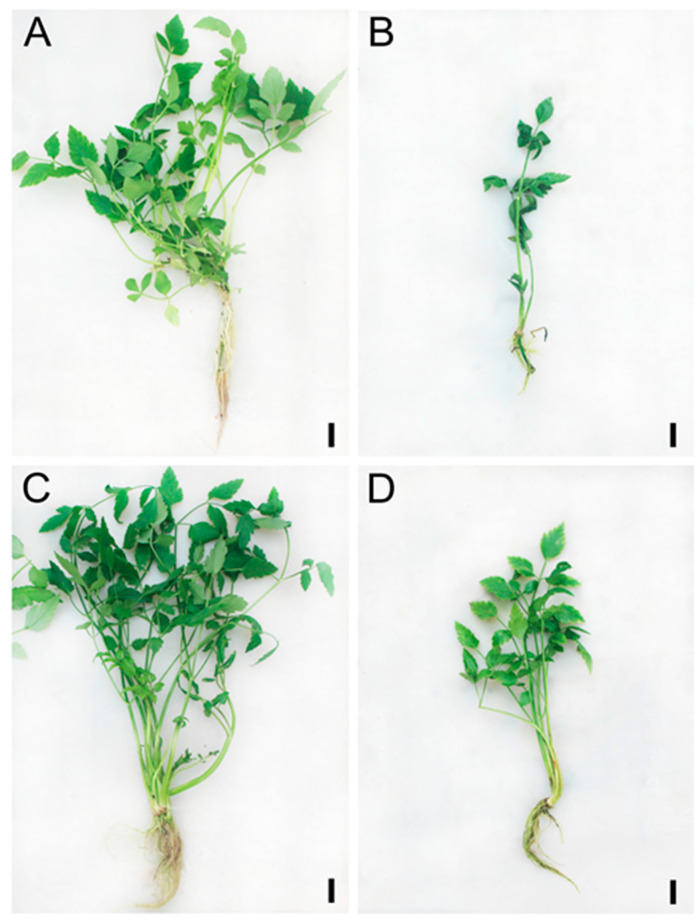
Effect of salt stress on phenotype of water dropwort (**A**) V11E0135 control (0 mmol/L NaCl), (**B**) V11E0135 treated (200 mmol/L NaCl), (**C**) V11E0022 control (0 mmol/L NaCl), and (**D**) V11E0022 treated (200 mmol/L NaCl). Bars = 1 cm.

**Figure 2 antioxidants-09-00940-f002:**
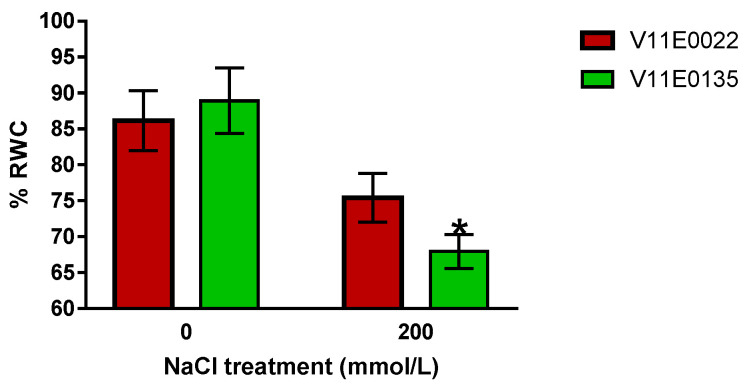
Relative water content (RWC) of water dropwort cultivars under salt stress. Bars with the asterisks (*) indicate the significant difference (*p* < 0.05) between the control and its respective treated samples. Values are means ± SD.

**Figure 3 antioxidants-09-00940-f003:**
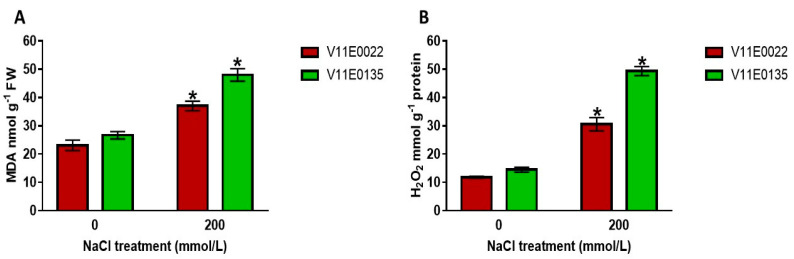
Effect of salt stress on malondialdehyde (MDA) (**A**) and reactive oxygen species (ROS) production (**B**) in water dropwort cultivars. Bars with the asterisks (*) indicate the significant difference (*p* < 0.05) between the control and its respective treated samples. Values are means ± SD.

**Figure 4 antioxidants-09-00940-f004:**
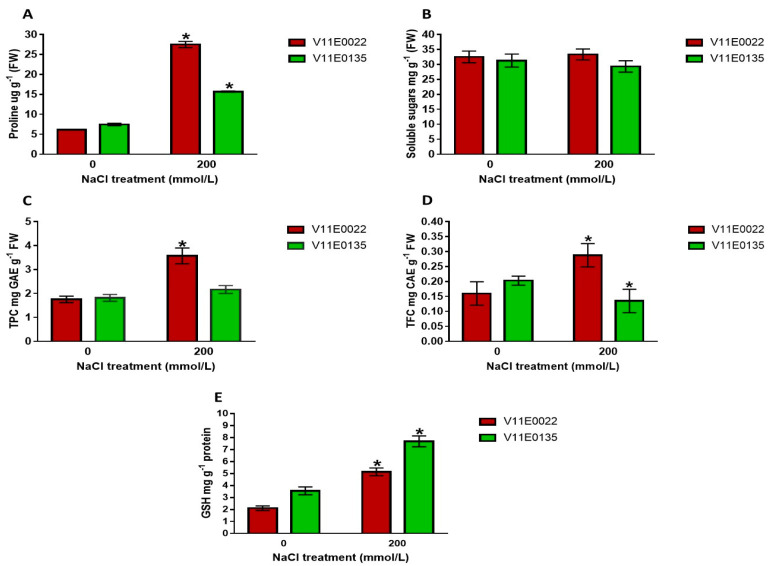
Effect of salt stress on the osmolytes and antioxidant molecules. (**A**) Content of proline, (**B**) content of soluble sugars, (**C**) total polyphenols compounds, (**D**) total flavonoids content, and (**E**) glutathione content in the water dropwort cultivars. Bars with the asterisks (*) indicate the significant difference (*p* < 0.05) between the control and its respective treated samples. Values are means ± SD.

**Figure 5 antioxidants-09-00940-f005:**
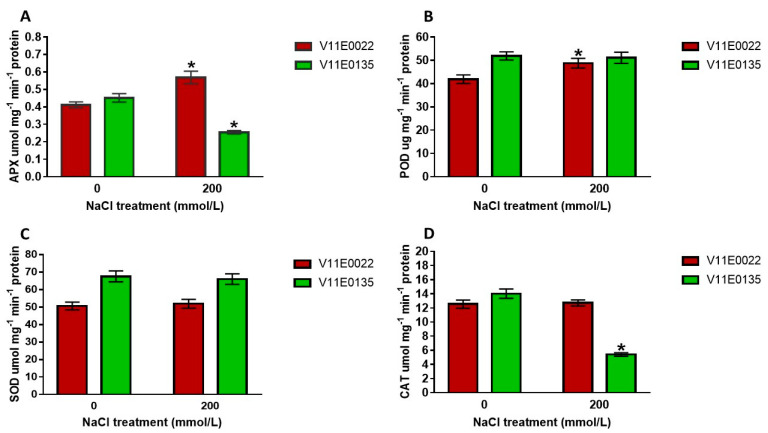
Effect of salt stress on antioxidant enzymes. (**A**) APX activity, (**B**) POD activity, (**C**) SOD activity, and (**D**) CAT activity in water dropwort cultivars. Bars with the asterisks (*) indicate the significant difference (*p* < 0.05) between the control and its respective treated samples. Values are means ± SD.

**Figure 6 antioxidants-09-00940-f006:**
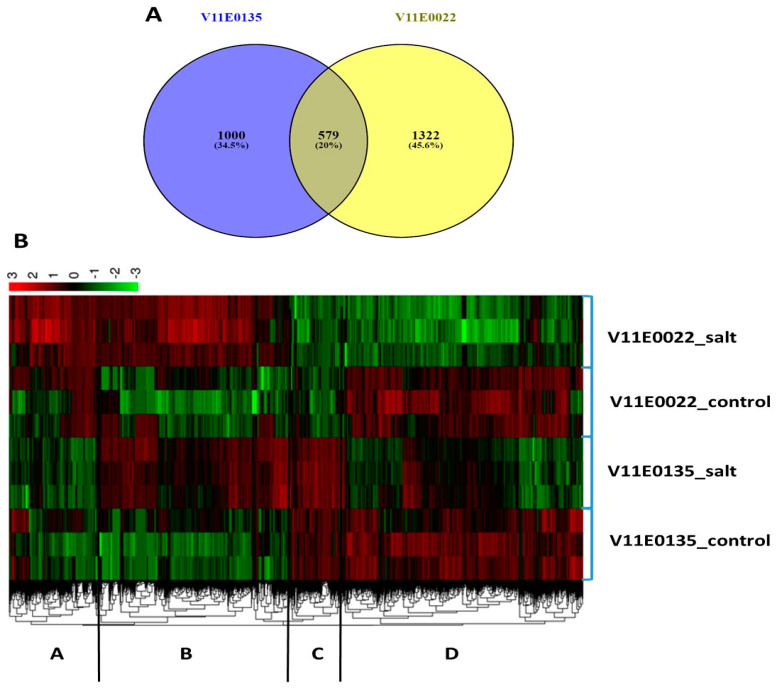
Distribution of differentially expressed genes (DEGs) in the leaves of both cultivars of the water dropwort under salt treatment. (**A**) Comparison of all DEGs (**B**) Cluster analysis of DEG profiles, data were divided into 4 data sets; V11E0135 control, V11E0135 salt treated, V11E0022 control, V11E0022 salt treated.

**Figure 7 antioxidants-09-00940-f007:**
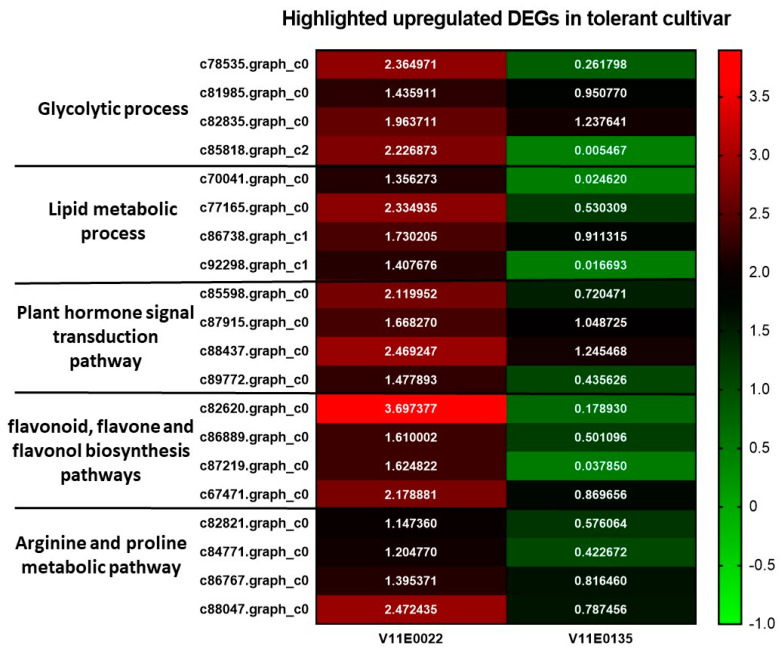
Heat maps of highlighted upregulated DEGs involved in different biological processes and pathways under salt stress conditions.

**Figure 8 antioxidants-09-00940-f008:**
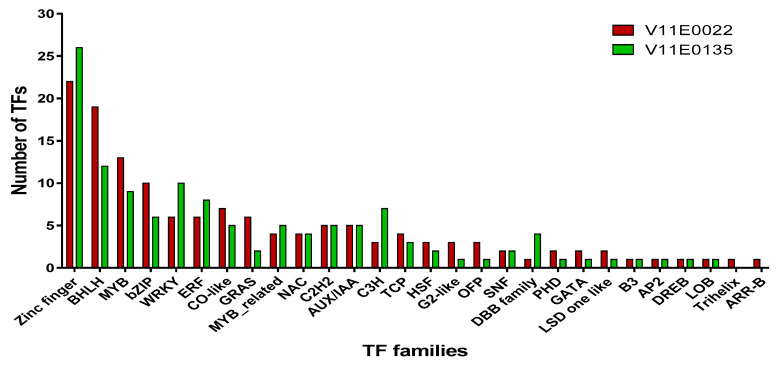
Distribution of differentially expressed transcription factors in two cultivars of water dropwort under salt stress.

**Figure 9 antioxidants-09-00940-f009:**
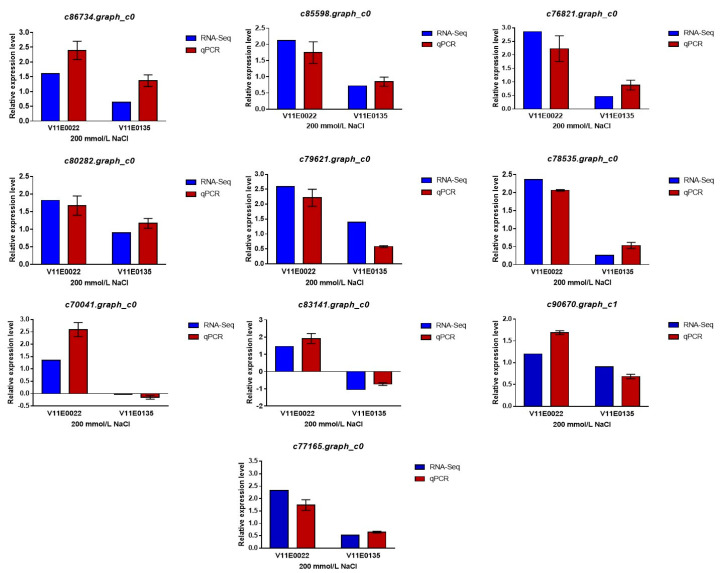
qRT-PCR analyses of DEGs responses of two water dropwort cultivars to salt stress. The expression patterns of 10 candidate genes were measured by qRT-PCR and RNA-seq under salt stress treatment. *Actin* was used as internal control. The individual black bars, representing the qPCR data, are the means ± SD.

**Table 1 antioxidants-09-00940-t001:** Effect of salt stress on phenotypic parameters and biomass of two cultivars of the water dropwort.

Cultivars	NaCl (mmol/L)	Plant Height (cm)	Root Length (cm)	Stem Length (cm)	Number of Branches	Number of Leaves	Shoot Fresh Weight	Root Fresh Weight
V11E0022	0	68.3 ± 1.5	23.0 ± 1.0	45.3 ± 0.6	8.0 ± 0.0	63.0 ± 3.0	50.93 ± 0.59	5.62 ± 0.35
	200	52.8 ± 1.3 *	16.5 ± 0.5 *	36.3 ± 1.5 *	5.0 ± 0.0 *	41.0 ± 1.7 *	27.85 ± 0.06 *	2.75 ± 0.21 *
V11E0135	0	58.3 ± 1.5	21.3 ± 0.6	37.0 ± 1.0	7.3 ± 0.6	64.3 ± 1.5	46.51 ± 1.07	4.83 ± 0.31
	200	40.0 ± 1.5 *	14.0 ± 0.5 *	25.7 ± 1.0 *	1.7 ± 0.6 *	9.7 ± 1.5 *	9.16 ± 0.36 *	1.37 ± 0.09 *

The asterisks (*) indicate a significant difference (*p* < 0.05) between the control and its respective treated samples. Values are means ± SD.

**Table 2 antioxidants-09-00940-t002:** Effect of salt stress on ionic content in two cultivars of the water dropwort.

Cultivars	NaCl (mmol/L)	Na^+^ (mg g^−1^ DW)	K^+^ (mg g^−1^ DW)
V11E0022	0	3.27 ± 0.15	70.93 ± 1.16
	200	33.57 ± 0.44 *	64.83 ± 1.04 *
V11E0135	0	4.47 ± 0.10	75.80 ± 0.96
	200	45.27 ± 1.22 *	63.59 ± 0.89 *

The asterisks (*) indicate a significant difference (*p* < 0.05) between the control and its respective treated samples. Values are means ± SD.

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
