# Peer review of "Investigation of an Antioxidative System for Salinity Tolerance in Oenanthe javanica"

_antioxidants, 2020, doi:10.3390/antiox9100940_

Round 1

Reviewer 1 Report

This manuscript addresses the defense mechanism against salinity stress in water dropwort. Two cultivars of dropwort are compared about their tolerance to salinity stress and antioxidant activity with or without the stress. This manuscript also has conducted transcriptome analysis revealing the differentially expressed genes between two cultivars.

The manuscript is well written, and the conclusion is supported by experimental evidence. This research will contribute to future work in this field. Therefore, I recommend the acceptance of this manuscript.

Author Response

Thank you for recommending the acceptance of our manuscript.

Reviewer 2 Report

Dear Sir or Madam,

the manuscript „Investigation of an antioxidative system for salinity tolerance in water dropwort“  addresses the question of inter-cultivar differences in terms of salt stress tolerance. The experiments are done at good experimental level, the methods are appropriately described and their selection is in agreement with the stated aims and conclusions. The data described in a straightforward way, although their presentation needs to be improved. The most problem of the manuscript is inappropriate quality of English. I am sure that the manuscript can be published in “Antioxidants” after an appropriate major revision.

Remarks

  1. The main problem of this manuscript is the language. Even worth – it is in the combination with wrong application of terms and definitions. For example – what is “the main stress parameter” (line 16)? In reality it is supposed to be “limiting environmental factor” here. The language quality is so low, that it affects the sense of the sentences. This fact dramatically affects the impression about the whole manuscript.Ans, by the way, salt, but not salinity stress is correct.
  2. The second big problem is the introduction – it indicates bad understanding of the mechanisms behind stress tolerance and adaptation. Indeed, how did the authors came to the idea that proline is an antioxidant? It is, actually, an osmoprotector, and the authors are supposed to know it, whem making and presenting such kind of study.
  3. The next problem is closely related: why the authors think, that “comprehensive comparisons between salt tolerant and sensitive cultivars are limited”? Just one of the fresh papers - Front. Plant Sci., 15 February 2019 | https://doi.org/10.3389/fpls.2019.00080. Did the authors see it and many others?
  4. The authors need to stick SI. And SI assumes mol/L as the units for molarity. “M” – is not an appropriate unit for this. Please, replace everywhere appropriately
  5. Figure 2- I find this art of presentation inconvenient – histogram or , better, box plot would be more appropriate.
  6. Figure 4. It is enough to give descriptions in the figure legends. There is no sense to write it in the panels additionally (e.g., see panel A- “proline concentration”)
  7. Figure 6 B- this form of presentation also makes no sense! It is clear, that no genes can be up- and down-regulated simultaneously.
  8. Figure 9: what is the sense to select genes randomly? Why not to address a specific group of genes – based on the results of functional annotation and pathway analysis?

Author Response

Thank you for your valuable suggestions and corrections.
The responses to comments are mentioned in the word file. Please find the attachment. Thanks 

Reviewer 3 Report

In the study by Kumar and colleagues (antioxidants-920004), two cultivars of the edible plant Oenanthe javanica were identified as tolerant [V11E0022, (Shaoyan shuiqin)] and sensitive [V11E0135, (Hefeizhongye shuiqin)] to salinity stress. On these cultivars phenotypic (growth, biomass, relative water content), physiological (lipid peroxidation, content in antioxidant compounds, antioxidant enzyme activity, ionic content) and high-throughput transcriptomic sequencing analysis have been carried out to shed light on the mechanisms underlying salt stress tolerance.

The study is well-conceived, well-written and it is, in my opinion, suitable for publication in Antioxidants. Small changes that could improve the quality of the manuscript are listed below.

In the title the scientific name of the plant species should be reported as the common name is little known outside the Asia.

Line 37:

Delete space after “in turn”

effect or “affect”?

Line 58:

Delete space after “studies on”

Lines 59-61: “…plants many aspects…”?

Line 67: “pharmacological activities against various ailments”?

Lines 84-85: “These studies can be of great importance to 85 understand salinity response mechanisms in O. javanica”

In my opinion it's a bit exaggerated.

Lines 93-95:

It is necessary to report the composition of Hoagland nutrient solution?

Line 224: “Previous reports mentioned that accumulation of NaCl in the cell wall and cytoplasm have negative effect on the plant length, number of branches and leaves in plants [45–49].”

Such a sentence is better suited in the Discussion section than in the Results.

Lines 231-233: “RWC is considered as a good and easiest parameter to check salt and drought stress [50]. Similarly, more decreased in the Relative Water Content (RWC) was found in V11E0135”

Replace with: “Relative Water Content (RWC) is considered as a good and easiest parameter to check salt and drought stress [50]. Similarly, more decreased in the RWC was found in V11E0135”

Figures 1, 2, 3, 4 and 5:

The meaning of letters should be explained in the figure legends. If I understand correctly, different letters represent statistically significant differences between control and treated samples. It would also be necessary to represent the significance of the differences between the two different cultivars under the same conditions.

Lines 248-249: “because antioxidants moleclues and antioxidant enzymes were always sharply induced to enhance tolerance in crops under abiotic stress conditions [8–11,51,52].”

Better suited in Discussion section

Line 251: “Salt tolerant cultivars with its predominant scavenging capacity…”

Replace with ““Salt tolerant cultivars with their predominant scavenging capacity…”

Lines 251 252: “Salt tolerant cultivars with its predominant scavenging capacity exhibit less lipid peroxidation and ROS production, such as H2O2 compared to their sensitive counterparts [53].”

Better suited in Discussion section

Lines 259-260: “Figure 3. Effect of salinity stress on ROS production and malondialdehyde (MDA) level in water dropwort cultivars.”

Replace with “Figure 3. Effect of salinity stress on malondialdehyde (MDA) level (A) and ROS production (B) in water dropwort cultivars. Different letters represent…”

Lines 267-268: “Similarly, flavonoids and soluble sugars were also increased in V11E0022, whereas, decline was observed in V11E0135, under salinity stress treatment (Figure 4B,D).”

Is sugar increase and decrease in V11E0022 and V11E0135 statistically significant? Is the increase in total phenols in V11E0135 statistically significant? This should be reported in both the text and in the figures.

Lines 277-279: “decrease was observed in V11E0135 in comparison to control, however, a significant decline was present in the activities of APX, and CAT (P < 0.05), under salinity stress treatment (Figure 5A,D).”

Check this sentence.

 Lines 284-285: “Yassin et al. in 2019 stated that the low uptake of Na+ and more uptake of K+ in higher plants are a main indicator for salt tolerance in higher plants [53]”

Better suited in Discussion section

Line 301:

Delete space between “and” and 140,405”

Lines 339-340: “Lipids as an important membrane component, which help plants to tolerate under abiotic stress conditions [54].”

Better suited in Discussion section

Lines 360-363: “Transcription factors (TFs) play an important role in regulation of the genes expression in response to various types of the abiotic stresses [18,55]. Numerous TFs including MYB, bZIP, WRKY, bHLH, ZF, NAC, and AP2/ERF families related to salinity stress have been previously identified in plants [17,55–61].”

Better suited in Discussion section

Lines 391-393: “This study proposed the tolerant cultivar based on its performance against salt stress, also suggested some physiological parameters as well as molecular mechanisms to evaluate the salt tolerance mechanism in water dropwort.”

Avoid repeating "mechanism"

Line 401: “…on the plants”

Replace with “on plants”

Line 569: “Proline”

lowercase letter

Author Response

Thank you for your valuable suggestions and corrections.

1: In the title the scientific name of the plant species should be reported as the common name is little known outside the Asia.

Response: Thank you for your valuable suggestion. We have replaced the common name with scientific name.
Lines 2-3: Investigation of an Antioxidative System for Salinity Tolerance in Oenanthe javanica

2: Line 37: Delete space after “in turn”

effect or “affect”?

Response: Thank you for correction.

Line 37: High concentration of salt leads to osmotic stress and ionic imbalance in plants, which in turn affect plant physiology.

3: Line 58: Delete space after “studies on”

Response: Thank you for correction. The space has been omitted.

Several studies on molecular level were also conducted, (Line 67).

4: Lines 59-61: “…plants many aspects…”?

Response: To date plants have been studied from different aspects, for example, some researchers have focused on TFs, antiporters, hormones etc. The description is elaborated as follow:
Lines 68-70: These studies have focused on the mechanisms underlying the salt tolerance in plants from different aspects such as transcription factors (TFs), plant hormones, antiporters, biosynthesis of secondary metabolites [24–26].

5: Line 67: “pharmacological activities against various ailments”?

Response: Thank you for the valuable comment. Changes are made as follows:

 Lines 73-77: Several studies have reported that water dropwort is a rich source of dietary fibers, starch, vitamins, and minerals, with excellent medicinal properties. Hyperoside, isorhamnetin, and persicarin are the key compounds present in water dropwort, which have pharmacological activities against different ailments such as hepatoprotective, anti-inflammatory, anti-arrhythmic, and antidiabetic.

6: Lines 84-85: “These studies can be of great importance to understand salinity response mechanisms in O. javanica”. In my opinion it's a bit exaggerated.

Response: Thank you for valuable suggestion. We have modified the sentence as follow:

Lines 94-95: These studies can help to understand salinity response mechanisms and identify genes of interest for breeding purpose in O. javanica.

7: Lines 93-95: It is necessary to report the composition of Hoagland nutrient solution?

Response: It is not necessary, but we thought that it might help readers or researchers interested to work in this area.

8: Line 224: “Previous reports mentioned that accumulation of NaCl in the cell wall and cytoplasm have negative effect on the plant length, number of branches and leaves in plants [45–49].” Such a sentence is better suited in the Discussion section than in the Results.

Response: Thank you for suggestion; we removed the mentioned sentences from results and relocated into discussion section.

Lines 416-417: It has been investigated widely that NaCl accumulation in the cell wall and cytoplasm can reduce plant length, the number of branches and leaves [58–62].

9: Lines 231-233: “RWC is considered as a good and easiest parameter to check salt and drought stress [50]. Similarly, more decreased in the Relative Water Content (RWC) was found in V11E0135”

Replace with: “Relative Water Content (RWC) is considered as a good and easiest parameter to check salt and drought stress [50]. Similarly, more decreased in the RWC was found in V11E0135”

Response: Thank you so much for suggestion. Following your suggestion, we have revised the description as follow:

Lines 256-258: Relative water content (RWC) is considered as a good and easiest parameter to check salt and drought stress [57]. Similarly, more decreased in the RWC was found in V11E0135 in comparison to the V11E0022 cultivar.

10: Figures 1, 2, 3, 4 and 5:

The meaning of letters should be explained in the figure legends. If I understand correctly, different letters represent statistically significant differences between control and treated samples. It would also be necessary to represent the significance of the differences between the two different cultivars under the same conditions.

Response: Thank you for the valuable suggestion. We have modified the graphs as per the suggestions from the Editor and Reviewer.

The asterisks (*) represent the significant difference (P < 0.05) between the control and its respective treated samples. Values are means ± SD.

Further, we have included explanation in the figure legends where applicable.

11: Lines 248-249: “because antioxidants moleclues and antioxidant enzymes were always sharply induced to enhance tolerance in crops under abiotic stress conditions [8–11,51,52].” Better suited in Discussion section.

Response: Thank you for valuable suggestion. We have moved the mention sentences to Discussion section.

Lines 439-441: To overcome the oxidative damage, plants have a defense system in the form of osmoprotectants, non-enzymatic compounds and antioxidant enzymes, which are always sharply induced under abiotic stress conditions [8,10,14,15,70,71].

12: Line 251: “Salt tolerant cultivars with its predominant scavenging capacity…”

Replace with ““Salt tolerant cultivars with their predominant scavenging capacity…”

Response: Thank you so much for the suggestion. Following your suggestion number 12 and 13, we have revised the description as follow:

Lines 432-433: Salt tolerant cultivars with their predominant scavenging capacity exhibit less lipid peroxidation and ROS production,

13: Lines 251 252: “Salt tolerant cultivars with its predominant scavenging capacity exhibit less lipid peroxidation and ROS production, such as H2O2 compared to their sensitive counterparts [53].” Better suited in Discussion section

Response: Thank you for suggestion; we removed the mentioned sentences from the results. These sentences are already mentioned in the discussion section.

Lines 432-433: Salt tolerant cultivars with their predominant scavenging capacity exhibit less lipid peroxidation and ROS production, such as H2O2 compared to their sensitive counterparts [68].

14: Lines 259-260: “Figure 3. Effect of salinity stress on ROS production and malondialdehyde (MDA) level in water dropwort cultivars.”

Replace with “Figure 3. Effect of salinity stress on malondialdehyde (MDA) level (A) and ROS production (B) in water dropwort cultivars. Different letters represent…”

Response: Thank you for correction. We have revised the description as follow:

Lines 283-285: Figure 3. Effect of salt stress on malondialdehyde (MDA) (A) and ROS production (B) in water dropwort cultivars. Bars with the asterisks (*) indicate the significant difference (P < 0.05) between the control and its respective treated samples. Values are means ± SD.

15: Lines 267-268: “Similarly, flavonoids and soluble sugars were also increased in V11E0022, whereas, decline was observed in V11E0135, under salinity stress treatment (Figure 4B,D).”

Is sugar increase and decrease in V11E0022 and V11E0135 statistically significant? Is the increase in total phenols in V11E0135 statistically significant? This should be reported in both the text and in the figures.

Response: Thank you for critical observation. In fact, sugars content was not significantly increased in V11E0022, whereas, an insignificant decline was observed in V11E0135 cultivar. Similarly, polyphenols were not significantly increased in V11E0135.
It is further clarified as follow:

Lines 289-295: Proline and total polyphenol concentrations were found higher in both cultivars under salt stress in comparison to the control (P < 0.05). However, V11E0022 showed a significantly higher level of proline, total polyphenolic compounds, and flavonoids compared to their counterpart (Figure 4A,C). Soluble sugars were also insignificantly increased in V11E0022 (Figure 4B). On the other hand, V11E0135 showed insignificant decline in soluble sugars and an insignificant increment in the concentration of polyphenols under salt stress compared to the control (Figure 4B,C).

16: Lines 277-279: “decrease was observed in V11E0135 in comparison to control, however, a significant decline was present in the activities of APX, and CAT (P < 0.05), under salinity stress treatment (Figure 5A,D).” Check this sentence.

Response: Thank you for suggestion. For better understanding, we have revised the description as follow:

Lines 303-308: Antioxidant enzymes (APX, SOD, POD, and CAT) were also found higher in V11E0022 under salt stress treatment. Notably, a significant increase was observed in the activities of APX and POD (P < 0.05) (Figure 5A,B). On the other hand, the activities of APX and CAT were significantly decreased in V11E0135 compared to the control (P < 0.05) (Figure 5A,D). Whereas, an insignificant decline was observed in the activities of POD and SOD in V11E0135 under salt stress treatment (Figure 5B,C).

17: Lines 284-285: “Yassin et al. in 2019 stated that the low uptake of Na+ and more uptake of K+ in higher plants are a main indicator for salt tolerance in higher plants [53]”

Better suited in Discussion section.

Response: Thank you for suggestion. We have moved this sentence to the discussion section.

Lines 565-567: Yassin et al. in 2019 stated that the low uptake of Na+ and more uptake of K+ in higher plants are the main indicator for salt tolerance in higher plants [68].

18: Line 301: Delete space between “and” and 140,405”

Response: Thank you for the observation.

Line 330: total of 58,126 unigenes and 140,405 transcripts

19: Lines 339-340: “Lipids as an important membrane component, which help plants to tolerate under abiotic stress conditions [54].” Better suited in Discussion section.

Response: Thank you for the suggestion. We have revised and moved this sentence to Discussion section.

Lines 489-490: Lipids are an important component of membranes, which assists in defense by acting as signal mediators to protect plants under biotic and abiotic stress [85].

20: Lines 360-363: “Transcription factors (TFs) play an important role in regulation of the genes expression in response to various types of the abiotic stresses [18,55]. Numerous TFs including MYB, bZIP, WRKY, bHLH, ZF, NAC, and AP2/ERF families related to salinity stress have been previously identified in plants [17,55–61].” Better suited in Discussion section.

Response: Thank you for the suggestion. We have revised and moved the description to Discussion section.

Lines 538-540: TFs play a crucial role in regulating genetic expression under abiotic stresses such as drought, salinity, and cold [103]. Numerous TFs including MYB, bZIP, WRKY, bHLH, ZF, NAC, and AP2/ERF families related to salt stress have been previously identified in plants [23,103–109].

21: Lines 391-393: “This study proposed the tolerant cultivar based on its performance against salt stress, also suggested some physiological parameters as well as molecular mechanisms to evaluate the salt tolerance mechanism in water dropwort.”

Avoid repeating "mechanism"

Response: Thank you for the correction. we adjusted as per your suggestion.

Line 411-413: This study proposed the tolerant cultivar based on its performance against salt stress, also suggested some physiological parameters as well as molecular mechanisms to evaluate the salt tolerance in water dropwort.

22: Line 401: “…on the plants” Replace with “on plants”

Response: Thank you for the correction. We have removed “the” as per your suggestion. (Line 421).

23: Line 569: “Proline” lowercase letter.

Response: Thank you for the correction. We have adjusted as per your suggestion. (Line 597).